# Modeling the acquisition shift between axial and sagittal MRI for diffusion superresolution to enable axial spine segmentation

**Robert Graf**[1,2]                                                                ROBERT.GRAF@TUM.DE
**Hendrik Möller**[1,2]                                                          HENDRIK.MOELLER@TUM.DE
**Julian McGinnis**[2,3]                                                        JULIAN.MCGINNIS@TUM.DE
**Sebastian Rühling**[1]                                                    SEBASTIAN.RUEHLING@TUM.DE
**Maren Weihrauch**[1]                                                    MAREN.WEIHRAUCH@TUM.DE
**Matan Atad**[1,2]                                                                MATAN.ATAD@TUM.DE
**Suprosanna Shit**[1,2]                                                      SUPROSANNA.SHIT@TUM.DE
**Bjoern Menze**[5]                                                              BJOERN.MENZE@UZH.CH
**Mark Mühlau**[3]                                                              MARK.MUEHLAU@TUM.DE
**Johannes C. Paetzold**[4]                                                    J.PAETZOLD@IC.AC.UK
**Daniel Rueckert**[2,3]                                                      DANIEL.RUECKERT@TUM.DE
**Jan S. Kirschke**[1]                                                          JAN.KIRSCHKE@TUM.DE

[1] *Department of Diagnostic and Interventional Neuroradiology, Klinikum rechts der Isar, TUM School of Medicine and Health, Germany*

[2] *Institut für KI und Informatik in der Medizin, Klinikum rechts der Isar, TUM School of Medicine and Health and School of Computation, Information and Technology, Germany*

[3] *Department of Neurology, Klinikum rechts der Isar, TUM School of Medicine and Health, Ger.*

[4] *Biomedical Image Analysis Group, Department of Computing, Imperial College London*

[5] *Department of Quantitative Biomedicine, University of Zurich, Switzerland*

**Editors:** Accepted for publication at MIDL 2024

## Abstract

Spine MRIs are usually acquired in highly anisotropic 2D axial or sagittal slices. Vertebra structures are not fully resolved in these images, and multi-image superresolution by aligning scans to pair them is difficult due to partial volume effects and inter-vertebral movement during acquisition. Hence, we propose an unpaired inpainting superresolution algorithm that extrapolates the missing spine structures. We generate synthetic training pairs by multiple degradation functions that model the data shift and acquisition errors between sagittal slices and sagittal views of axial images. Our method employs modeling of the k-space point spread function and the interslice gap. Further, we imitate different MR acquisition challenges like histogram shifts, bias fields, interlace movement artifacts, Gaussian noise, and blur. This enables the training of diffusion-based superresolution models on scaling factors larger than $6\times$ without real paired data. The low z-resolution in axial images prevents existing approaches from separating individual vertebrae instances. By applying this superresolution model to the z-dimension, we can generate images that allow a pre-trained segmentation model to distinguish between vertebrae and enable automatic segmentation and processing of axial images. We experimentally benchmark our method and show that diffusion-based superresolution outperforms state-of-the-art super-resolution models.

**Keywords:** Superresolution, MRI, Spine, Denoising Diffusion, Segmentation, Degradation function, MR Preprocessing

## 1. Introduction

Magnetic Resonance Imaging (MRI) has become a cornerstone of medical imaging. While improvements in MRI scanners have enabled widespread clinical adaptation, imaging requires a careful balance of acquisition time, required scan resolution, and signal quality (Plenge et al., 2012). To reduce scan time, medical imaging research has developed a variety of super-resolution (SR) algorithms that aim to reconstruct high-resolution from low-resolution scans featuring sharp and anatomically faithful details whilst ideally being resistant to hallucinations. While advancements have led to noticeable achievements in various MRI applications, improving spinal multiple sclerosis (MS) imaging is particularly challenging. In MS diagnosis, precise imaging is required, as detecting MS lesions is inherent to diagnosing and monitoring the disease (Wattjes et al., 2021; Kearney et al., 2015). Registration and structure/MS segmentation are well established in brain imaging, while spinal imaging has been trailing behind due to its more challenging acquisition and analysis process. Only recently, it gained traction within the MS community since guidelines have shifted a focus to spinal cord lesions (Wattjes et al., 2021) and recent technical advances such as the development of the Spinal Cord Toolbox (SCT) (Leener et al., 2017). In clinical routine, T2-weighted (T2w) MRI is crucial in assessing lesion load and disease activity. Due to the size of the spine, imaging is typically performed using 2D sagittal or axial views with relatively high slice thicknesses, typically ranging between 3-10 mm. This poses an inherent challenge to 3D segmentation, as axial 2D views frequently lack the intricate details and context needed to accurately segment vertebral structures and intervertebral discs. At the

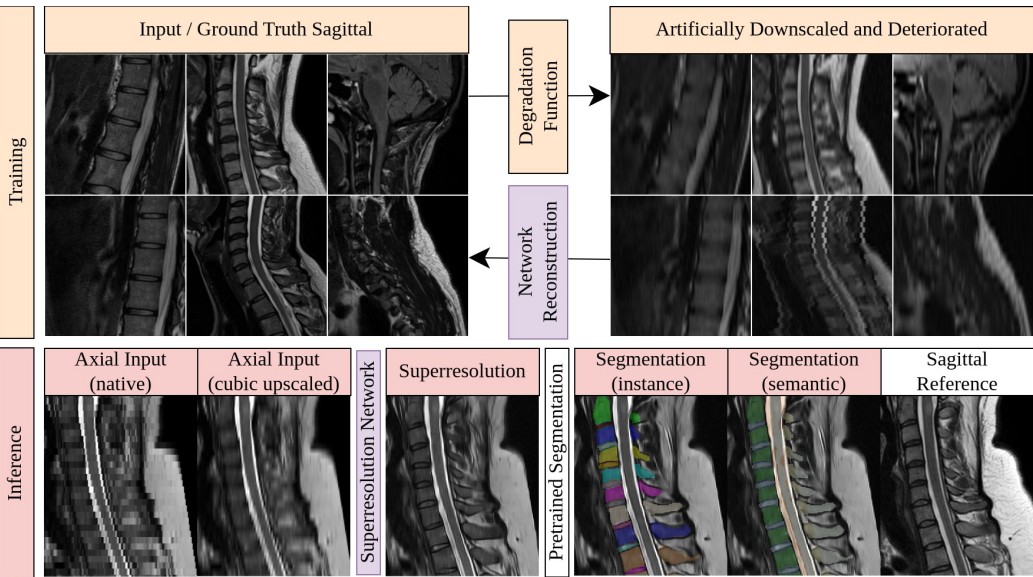

Figure 1: Training procedure (yellow): We take real sagittal slices, apply our degradation function, and train a diffusion model to reconstruct the image. Inference (red): 1) the axial image is upscaled to the target resolution; 2) the sagittal slices of the axial images are super-resolved; 3) sagittal segmentation algorithms succeed in vertebrae segmentation.

same time, axial scans are preferred in the context of lesion activity. To conduct MS studies for the spinal cord, it is necessary to register spine scans to a common template like the PAM50 atlas (De Leener et al., 2018). Current registration tools (Leener et al., 2017) require manual vertebra-level labels to function. Although whole spine segmentation for sagittal images was made available recently (Graf et al., 2023; Möller et al., 2024), the problem of axial vertebra instance segmentation persists due to the challenges posed by large slice thicknesses. Through super-resolution, we can enable segmentation on axial images. With such a segmentation, we could automatically extract the vertebra-level label, which can be utilized for atlas registration and enables large-scale studies to access MS lesion growth in the spinal cord. This paper proposes a solution to these challenges by enhancing the quality of axial images for segmentation to the same resolution as the inplane resolution of a sagittal slice. The approach involves using sagittal images only for training, avoiding the difficulties associated with aligning sagittal and axial images. We improve the existing degradation model of SMORE (Zhao et al., 2021), based on MRI scan properties, to learn the impact of shifts in distribution between axial and sagittal images. We change the zero-shot-learning approach of SMORE to an unpaired sagittal plane of axial to sagittal super-resolution. We compare existing super-resolution networks with the conditional diffusion models Palette. Our pipeline is the first that can automatically segment full vertebrae in axial T2w images. We achieve a realistic-looking >6x super-resolution without paired or isotropic data.

## 2. Related Work

**Superresolution.** Superresolution (SR) enhances image sharpness and details. Various models have been introduced to upscale paired low-resolution (LR) images to high-resolution (HR) images (Bashir et al., 2021; Bhowmik et al., 2017). In the context of MRI, isotropic HR images are often downscaled to LR images through downsampling for training (Feng et al., 2022; Chen et al., 2018; Li et al., 2022). Subsequently, a deep learning model, such as a GAN (Zhou et al., 2022) or regression model, learns the inverse process. More recent papers investigate denoising diffusion as an SR model (Chung et al., 2023; Wu et al., 2023; Kawar et al., 2022). Some SR methods focus on enhancing LR patches to HR patches (Mithra et al., 2021; Zhao et al., 2021). For instance, SMORE (Zhao et al., 2021) utilizes zero-shot-learning, where high-resolution plane patches are degraded with an MRI-specific point spread function. The orthogonal LR plane is then upsampled and super-resolved by the model. However, a limitation of patch-based SR is its inability to introduce features that are larger than a patch or perform biologically informed slice inpainting, which is crucial for our task because we want to reintroduce the z-axis information to distinguish vertebra instances. The SMORE zero-shot approach can not learn how a vertebra looks under different rotations because it has never seen it, especially given the large-scale factors. Other papers explore jointly modeling axial and sagittal images (Liu et al., 2021; McGinnis et al., 2023; Zhao et al., 2019), aiming to enhance correctness. Nevertheless, these approaches necessitate registration to align these images, posing a "chicken and egg" problem for our task.

**Spine segmentation.** Vertebra segmentation is well-established for CT through the VerSe Challenge (Sekuboyina et al., 2021) and its participants (Payer et al., 2020; Chen et al., 2020). Regarding sagittal T2w MRI, two datasets are available for the lumbar spine—Spider (van der Graaf et al., 2023) and MRSpineSeg Challenge (Pang et al., 2020).

Li et al. (2021) developed a 2D slice-wise semantic segmentation for axial images. Recent advancements have extended CT whole spine segmentation to sagittal MR images (Graf et al., 2023). Möller et al. (2024) leveraged these labels and incorporated intervertebral disc and spinal channel segmentation from Streckenbach et al. (2022). We employ their publicly available segmentation algorithm as a downstream task on our super-resolved axial images to generate segmentation. On native axial resolution, this model can not untangle vertebra instances due to the strong resolution mismatch.

**Registration.** The Spinal Cord Toolbox (Leener et al., 2017) facilitates registration to an atlas when spinal cord segmentation and vertebra height points are accessible. However, automatic vertebra height extraction proves unreliable on thick axial images and necessitates manual annotation. In contrast, our segmentation on axial SR can autonomously generate these points, offering a fully automatic solution.

## 3. Methodology

**Problem statement** Given two types of slice-wise MRI with different high-resolution planes, we want to upscale a low-resolution view to the same resolution as the other image. For this, we aimed to investigate degradation functions and a diffusion-based SR algorithm. Pairing real data is infeasible due to the thick sliced spine images suffering from intrinsic motion, missing isotropic ground truth data, and misaligned sagittal volumes. To assess the quality of the SR data, we assessed spine segmentation as a downstream task. The segmentation enables labeling and registration to a common atlas to evaluate multiple sclerosis statistics in the axial images in future studies, a task previously not solved.

**Dataset** We have an internal dataset with 416 multiple sclerosis patients (split 336/45/36) with multiple follow-up sessions (total=927/136/93). Each session contains $\approx 2-4$ axial (total=2742/404/282) and $\approx 2$ sagittal T2w MRI scans (total=1883/285/193). Our data exhibits a through-plane spacing of 3-4 mm in sagittal images and 5 mm in axial images, representing a standard slice thickness for such scans. See Figure 1 on how the resolution differs between sagittal views of an axial and a sagittal image. Due to the large partial volume effects and inter-vertebral movement, we could not register the axial and sagittal image beyond the automatic alignment with an error of up to 30 mm (Figure 7).

### 3.1. Degradation function

A degradation function is the forward mapping from an HR to an LR image. A naive approach would be interpolation and random Gaussian noise (Bashir et al., 2021). For MRI images, this approach does not correctly model the signal acquisition.

**Point spread function (PSF)** MRIs are acquired in k-space, and the PSF must be modeled in k-space instead of the image space. Inspired by SMORE (Zhao et al., 2021), we use a convolutional kernel to degrade the 2D HR image that simulates the partial volume effect caused by the k-space PSF. While SMORE only downscales the image to LR, we both **down- and upscale** the image to the same pixel space as the HR image. We add random Gaussian noise in between the up- and downsampling process. The target sagittal in-plane resolution is the same as the pretrainend segmentation network with 0.8571 mm.

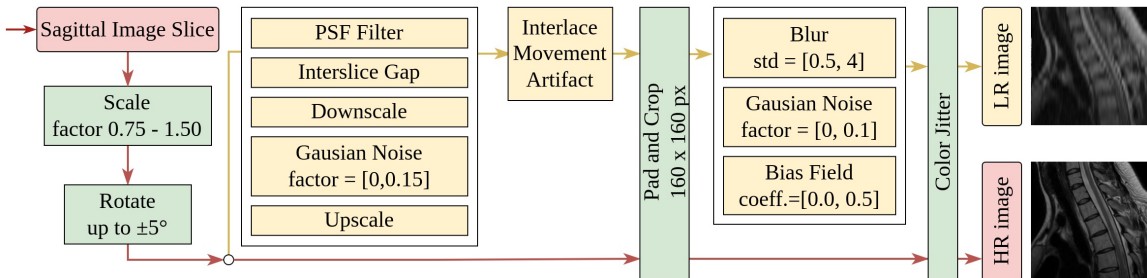

Figure 2: Overview of our degradation functions. We start with a sagittal image slice. The red path describes the augmentation of the HR image and the yellow path for the LR image. The final LR and HR images constitute the training pairs for the SR model.

We work with clinical data and expect that we cannot rely on the resolution being always exactly 5 mm. With the down- and upscale steps, we are not limited to any fixed integer scaling factor. Instead of one fixed kernel, we take **multiple PSF kernels** with starting resolutions between 4 and 9 mm in 0.25 mm steps.

**Interslice gap randomization** The signal of a 2D slice is measured with a target thickness. The thickness of this volume is not the slice distance (Bradley and Glenn, 1987; Schwaighofer et al., 1989). We found the measured thickness to be 4 mm. This volume is not constantly integrated but must drop from the center focus to the 2 mm edges to the side; otherwise, we would observe very blurry images. As we only have discrete pixel rows, we take up to half (1 to $n$ slices chosen randomly) of them inside a slice thickness and weighted sum them together. We repeat this generated pixel row for the full slice thickness. This process simulates a random volume effect rather than the real partial volume effect and intentionally loses information about slices between two axial image slices. This process is done before the downsampling of the SMORE degradation.

**Histogram** MRI images have no fixed values but are instead images between 0 and a max-value. We linearly rescaled the image to [0,1]. The histogram changes between scans and MRI devices. We used color jitter to randomize the histogram (0.8 - 1.2 factor). This should make the model histogram shift agnostic. We tried manual parametric histogram shifts, but they added nothing to the performance when we used them, along with the more general color jitter.

**Interlace movements artifact** Slice interleaving is the process of scanning every other slice first to prevent cross-talk between slices. If the subject moved during the image acquisition, we observe that every other slice is shifted in the front-back direction. We simulate this in 15% of the LR images by moving an interlace pattern with 4 to 10 pixels thickness by 0.2% to 2% of the image width to the back. See Figure 4 for an example.

**Others** We add other common image degradations like random MRI-bias field (Van Leemput et al., 1999; Sudre et al., 2017), Gaussian noise, and blur. We do a random rescale (factor 0.75 to 1.5) and randomly rotate up to 5 degrees. PSF and interslice gab simulation are applied all the time. All other degradation have a chance of 30% to be applied. For the

order of operation, see Figure 2. We pad and crop the images to 160 by 160, which is roughly the size of a sagittal view of an axial image.

### 3.2. Conditional diffusion model for superresolution

Our chosen SR-model is Palette, a diffusion image to image model (Saharia et al., 2022). Palette is a diffusion model for conditional image translation, inpainting, and image reconstruction. It use the denoising diffusion implicit model (Song et al., 2020; Ho et al., 2020), which is conditioned by concatenating the LR image to the noised HR image as an input. The model is a 4-block UNet. It uses SiLu activation and 32-channel GroupNorm. The timestep embedding is added by scaling and shifting the channels in every ResBlock. We only superresolve single sagittal slices. The slice superresolution is mostly consistent, but some image artifacts are visible in the axial view. Like Zhao et al. (2021), we could introduce a second phase that removes the created artifact from the axial view by superresolving an axial image and producing a paired data set of the original image and its corresponding slices. We omitted this for simplicity.

## 4. Experiments and Results

We manually corrected a vertebra instance segmentation ground truth for our axial test data, where at least the vertebral body is fully segmented. Predicted vertebra instances are matched with the center of mass of the vertebra body to this ground truth. We count how many vertebrae are missed by merging or omitting. A vertebra is considered omitted when no vertebra body center mass point is inside the instance-corrected ground truth mask. We analyze the vertebra components by computing the Betti error $b_0$ and $b_1$ (Stucki et al., 2023). We expect a vertebra to be a single component $b_0 = 1$. A vertebra has a single hole for the spinal cord, and the neck vertebrae contain two additional holes called transverse foramina, which are openings for vessels to pass through. The used segmentation algorithm cannot reproduce the additional holes. We expect the number of holes to be $b_1 \in [1, 2, 3]$ for neck vertebrae and $b_1 = 1$ for the rest. We count error rates per vertebrae that are missing or do not fit this description. We use this metric because we observed that the segmentation network often has issues with completing the structure around the spinal channel. We manually annotate 107 random axial slices of the bony vertebra, each in a different axial volume, by two medical experts. The sagittal segmentation algorithm segments the superresolved axial image. Then, we sample the segmentation back to the original volume and compute the Dice score for our expert annotation. Unlike other papers, we omitted PSNR or SSIM from our analysis due to the lack of a suitable ground truth. As a proxy, we use a downstream task quality to evaluate whether contrast enhancement and inpainting improve the segmentation and instance detection. Therefore, this work optimizes the structural enhancement instead of the perceptual quality of the images.

### 4.1. Model comparison

We compare existing models with the Palette diffusion approach, all with our degradation function. We use cubic interpolation, GAN-based ESRGAN+RRDBNet (Wang et al., 2018; Feng et al., 2022), regression-based RCAN (Zhang et al., 2018) and transformer-based

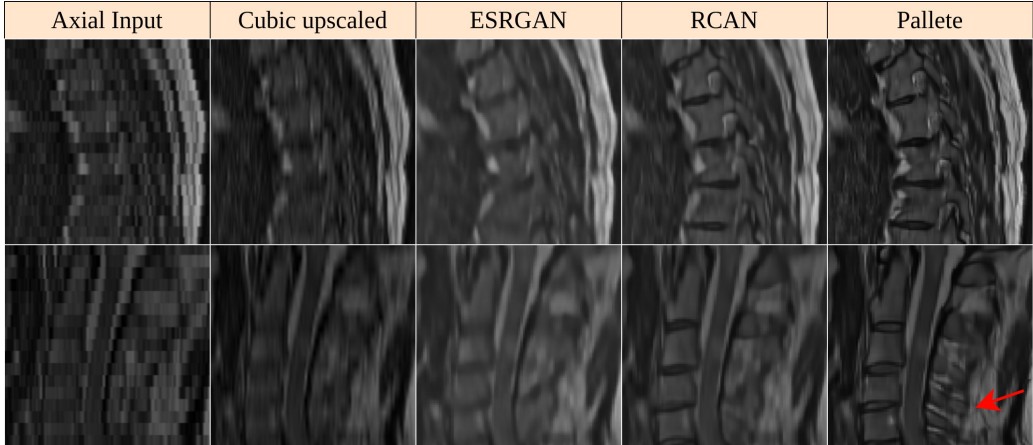

| Axial Input | Cubic upscaled | ESRGAN | RCAN | Pallete |
|---|---|---|---|---|

Figure 3: Qualitative Results: Model differences for two samples. All models improve image detail, but the contrast strength between bone and soft tissue is model-dependent. Only our model (Pallete) can reconstruct all posterior neck vertebra structures. See red arrow.

HAT(Chen et al., 2023) model as baselines. We report them in Table 1 and visualize differences in Figure 3. The diffusion model Palette was better suited than the best baseline RCAN for superresolution. RCAN performance is competitive with Palette. As already discussed by Zhang et al. (2018), all their tested SR models drop in performance with large scaling factors. We reckon Pallete could handle larger superresolution factors better than other models. Further research is needed for confirmation.

Table 1: Quantitative comparison of downstream 3D segmentation. The axial SR is segmented by a pre-trained network. Vertebra detection rate (dtr) is the percent of vertebrae correctly identified through translation and segmentation in axial images. We reported the Betti numbers error rate (er) when a vertebra has not $b_0 = 1$; $b_1 = 1$ or $\in [1, 2, 3]$ for neck vertebra. "axial Dice" is the Dice score on a binary vertebra-bone annotation with 107 slices in different axial scans evenly distributed through the subjects.

|  | Vertebra dtr↑ | Betti $b_0$ er↓ | Betti $b_1$ er↓ | axial Dice↑ |
|---|---|---|---|---|
| cubic interpolation | 0.6464 | 0.435 | 0.600 | 0.684 |
| ESRGAN (RRDBNet) | 0.9224 | 0.141 | 0.318 | 0.695 |
| HAT 4×4 blocks | 0.9805 | 0.098 | 0.240 | 0.681 |
| RCAN | 0.9853 | 0.062 | 0.165 | 0.711 |
| Palette (diffusion) | **0.9942** | **0.045** | **0.115** | **0.718** |

## 4.2. Degradation function ablation

We do an ablation on our degradation function. See Table 2 for exact values. The Dice score of axial slices does not improve much because of large interpolation artifacts. The segmentation network rescales the image in the left/right direction to 1.25 mm. Other common errors are structures not modeled in the sagittal segmentation, like the transverse

Table 2: Ablation of the degradation function. We retrained the model on different degradation aspects, as indicated on the table's left side. The right side has the same metrics as before.

| Color Jitter | Down- and Upscale | SMORE one PSF | SMORE multi PSF | Interslice gap | Movements artifact | Rescale, Rotate | Blur, BiasField,Noise | Vertebra dtr↑ | Betti $b_0$ er↓ | Betti $b_1$ er↓ | axial Dice ↑ |
|---|---|---|---|---|---|---|---|---|---|---|---|
| | | | cubic upscale | | | | | 0.6464 | 0.435 | 0.600 | 0.684 |
| ✓ | ✓ | | | | | | | 0.9851 | 0.053 | 0.153 | 0.710 |
| ✓ | ✓ | ✓ | | | | | | 0.9937 | 0.043 | 0.142 | 0.713 |
| ✓ | ✓ | | ✓ | | | | | 0.9931 | 0.049 | 0.133 | 0.717 |
| ✓ | ✓ | | ✓ | ✓ | | | | 0.9931 | 0.047 | 0.118 | 0.712 |
| ✓ | ✓ | | ✓ | ✓ | ✓ | | | 0.9937 | 0.043 | 0.124 | 0.705 |
| ✓ | ✓ | | ✓ | ✓ | ✓ | ✓ | | 0.9937 | **0.039** | 0.133 | 0.714 |
| ✓ | ✓ | | ✓ | ✓ | ✓ | ✓ | ✓ | **0.9942** | 0.045 | **0.115** | **0.718** |

foramina or the transverse process of the vertebra, which is outside of the field of view of most sagittal images. Our approach improves the vertical correctness of the images. The betti errors towards $b_0$ do not follow a trend, while $b_1$ shows our additional augmentations improve the segmentation around the spinal channel. Our ablation is limited to additional augmentations because the chosen SR-model plus segmentation model is quite stable, even with some SR anomalies, and we believe that we have reached the segmentation model's upper limit. We introduced additional augmentation to be invariant against movement artifacts, MR bias-field, changing through-plane resolution, and image corruption through noise or blur. The test showed that these changes do not worsen the baseline results.

## 5. Conclusion

In this work, we address the previously unsolved task of full 3D vertebrae instance segmentation in clinical axial MRI images. This was unsolved because the z-resolution of clinical axial MRI images is typically larger than the intervertebral disc or substructures of the vertebrae. We achieve this by training unpaired superresolution diffusion models with a newly developed degradation function. Our degradation function is informed by real-world MRI properties such as the point spread function, the interslice gap, the interlaced acquisition, histogram shift, and bias fields. We can artificially introduce a domain shift from HR to LR images and create aligned training data. We do not require isometric data nor aligned axial and sagittal images. Our diffusion model can inverse this process by training on the generated pairs. During inference, we only need the real axial images to superresolve them. SR for large scaling factors is strongly ill-posed, and our model is limited to a best guess. The structures between slices, as expected, vary from the real structure. The superresolution enables us to segment the axial image and compute points for registration; both were previously not possible. See Figure 5 and 6. In the future, our approach can be extended to any other MR acquisition protocol with resolution differences.

## Acknowledgments

The research for this article received funding from the European Research Council (ERC) under the European Union's Horizon 2020 research and innovation program (101045128 — iBack-epic—ERC2021-COG). JM, MM and JSK are supported by the Bavarian State Ministry for Science and Art(Collaborative Bilateral Research Program Bavaria – Québec: AI in medicine, grant F.4-V0134.K5.1/86/34). SR has received funding from the Technical University of Munich, TUM School of Medicine and Health, Clinician Scientist Programme (KKF), project reference H-09.

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

## Appendix A. Supplementary Material

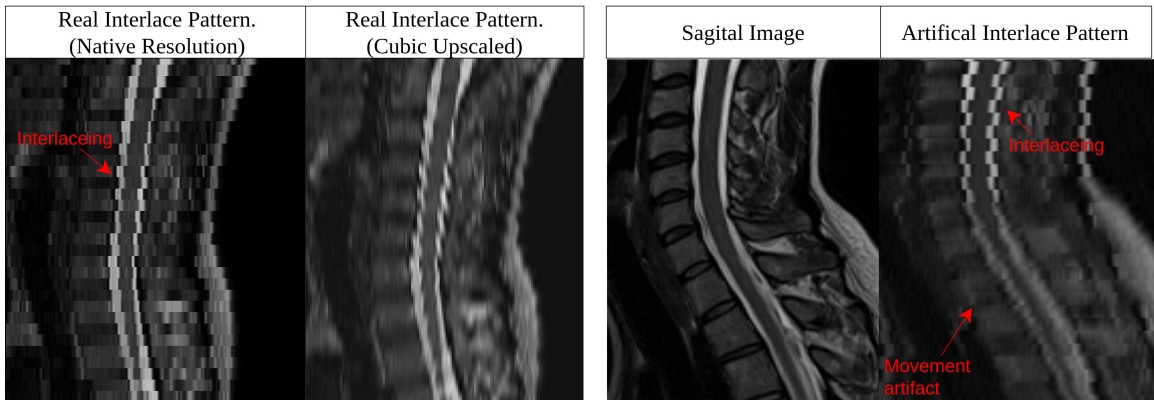

Figure 4: Real and artificial interlace pattern. It is easy to spot when the pattern is orthogonal to the spinal cord. The introduced movement artifacts are difficult to spot in other areas. The artificial interlace pattern has the SMORE degradation and the interlace pattern; The other degradation functions are turned off.

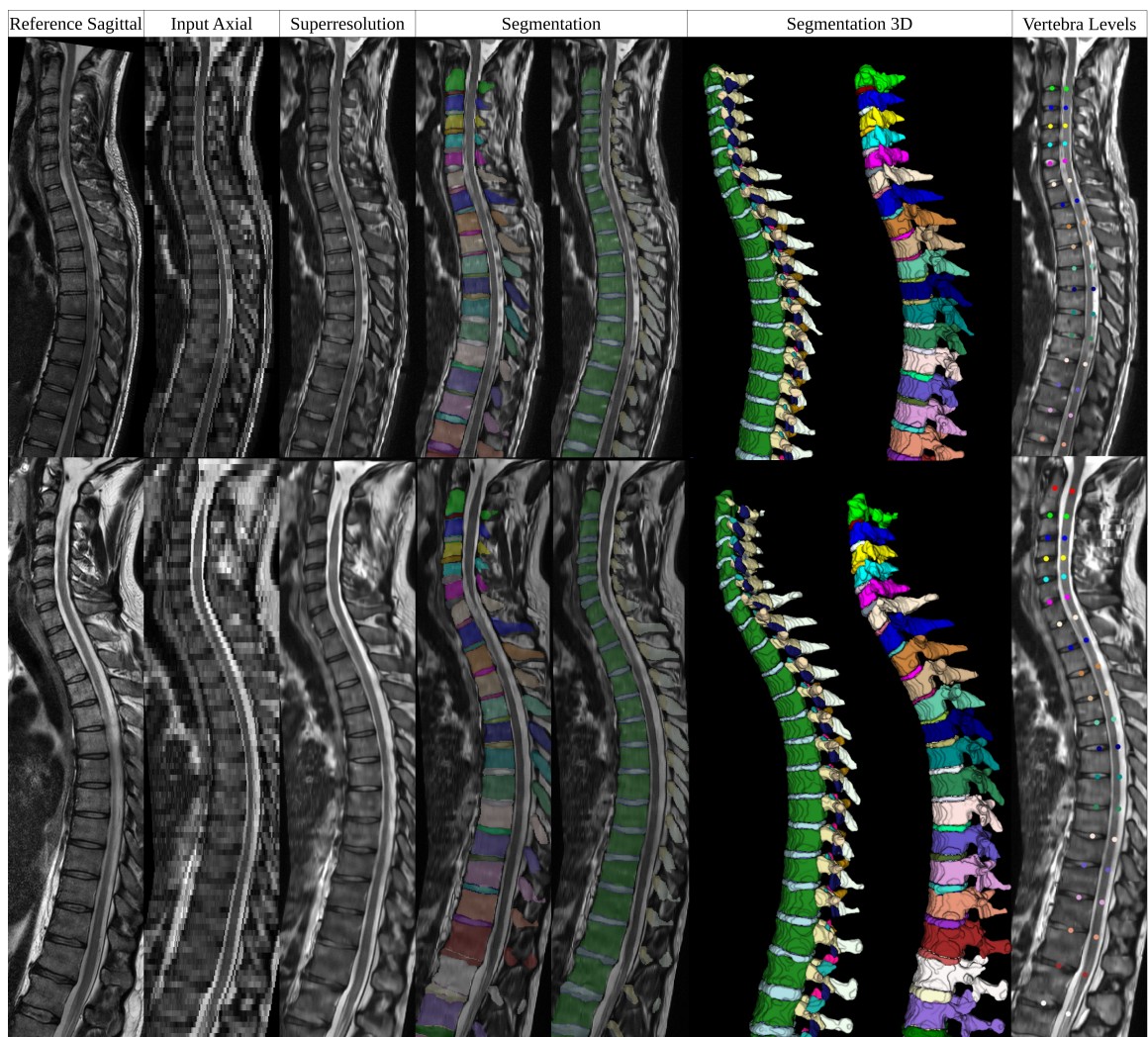

| Reference Sagittal | Input Axial | Superresolution | Segmentation | Segmentation 3D | Vertebra Levels |

Figure 5: Examples of our inference pipeline. We stitched and cubic upscaled the axial images to isometric 0.8571 and used the sagittal plane to upscale the image. We show the segmentation in the slice and 3D render. The segmentation can be used to extract the vertebra heights for registration.

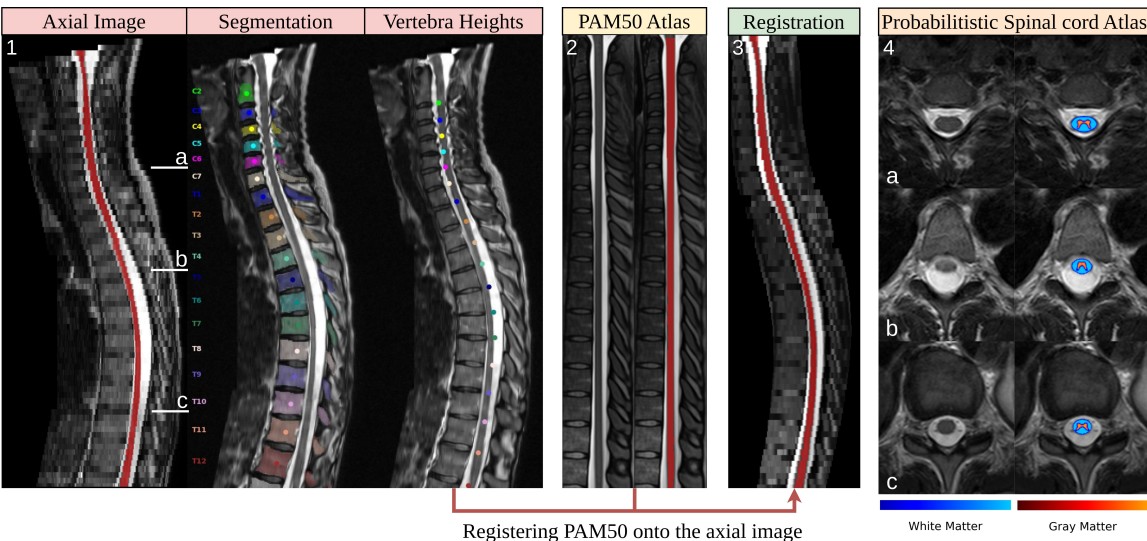

Figure 6: An example of an atlas registration with our automatic vertebra level points. Block 1 is the axial image with superresolution, automatic segmentation, and vertebra heights. Block 2 is the PAM50 atlas with and without spinal cord segmentation. Block 3 shows PAM50 registered on the axial image. Block 4 shows axial images with the probabilistic white and gray matter atlas registration from the PAM50 template. The height where those slices are taken is indicated with a,b,c in block 1.

Table 3: We repeated the test on an older version of the segmentation algorithm, which is more susceptible to image anomalies. There, we observe that the final degradation improves the vertebra detection rate. The Betti error rates behave similarly, but $b_1$ is much worse overall.

Ablation of the degradation function with an older version of the segmentation model that is less stable towards out-of-distribution images and performs worse in the neck region. We retrained the model and turned on different degeneration, as indicated on the table's left side. The models are the same as in the other tables. Vertebra detection rate (dtr) is the percent of vertebrae correctly identified through translation and segmentation in axial images. We reported the Betti numbers error rate (er) when a vertebra has not $b_0 = 1$ $b_1 = 1$ or $\in [1, 2, 3]$ for neck vertebra. "axial Dice" is the Dice score on a binary vertebra-bone annotation with 107 slices in different axial scans evenly distributed through the subjects.

| Color Jitter | Down- and Upscale | SMORE one PSF | SMORE multi PSF | Interslice gap | Movements artifact | Rescale, Rotate | Blur, BiasField,Noise | Vertebra dtr↑ | Betti $b_0$ er↓ | Betti $b_1$ er↓ | axial Dice ↑ |
|---|---|---|---|---|---|---|---|---|---|---|---|
| \multicolumn{8}{c}{cubic upscale} | | | | | | | | 0.7557 | 0.495 | 0.756 | 0.599 |
| ✓ | ✓ | | | | | | | 0.9702 | 0.160 | 0.446 | 0.663 |
| ✓ | ✓ | ✓ | | | | | | 0.9810 | 0.155 | 0.435 | 0.661 |
| ✓ | ✓ | | ✓ | | | | | 0.9773 | 0.179 | **0.389** | 0.664 |
| ✓ | ✓ | | ✓ | ✓ | | | | 0.9795 | 0.155 | 0.431 | **0.665** |
| ✓ | ✓ | | ✓ | ✓ | ✓ | | | 0.9831 | **0.133** | 0.408 | 0.655 |
| ✓ | ✓ | | ✓ | ✓ | ✓ | ✓ | | 0.9926 | 0.134 | 0.410 | 0.660 |
| ✓ | ✓ | | ✓ | ✓ | ✓ | ✓ | ✓ | **0.9932** | 0.152 | 0.413 | 0.659 |

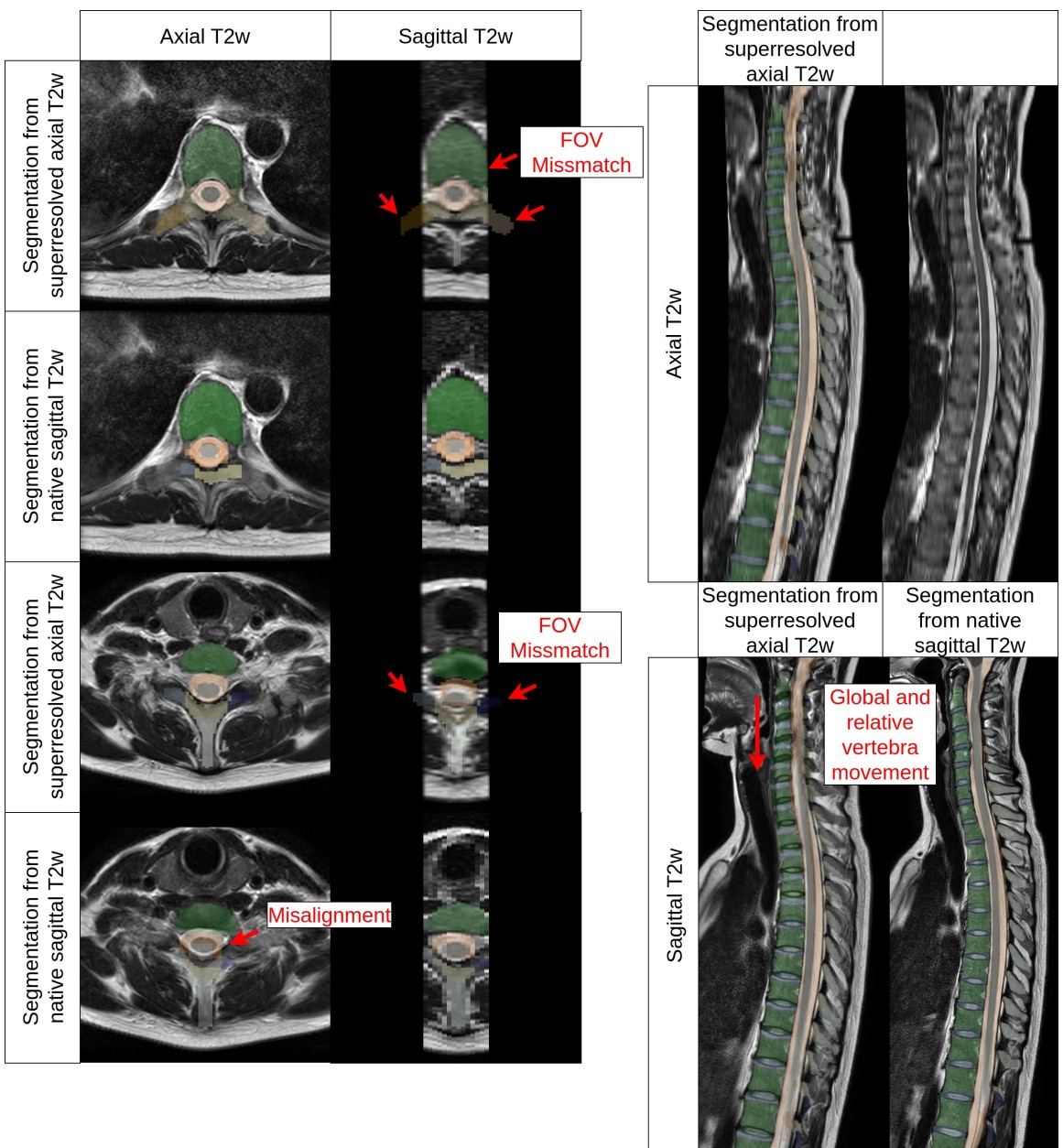

Figure 7: Example of how axial and sagittal T2w scans may cover different fields of view (FOV) and may include localized mismatches in anatomical structures. Matching the structures would require deformable registration; this does not routinely work without strong supervision due to different fields of view and subtle structural differences caused by highly anisotropic imaging. The shown segmentation for axial images was unavailable before and could not be used for registration because it is an end product of our superresolution method, plus the sagittal segmentation algorithm. Our SR model solved the chicken and egg problem by not requiring registration.

