# OpenReview forum: "Modeling the acquisition shift between axial and sagittal MRI for diffusion superresolution to enable axial spine segmentation"
_MIDL.io/2024/Conference — MIDL 2024 Poster_

### Official Review · Reviewer_HXjK · 2024-02-22

**Confidence:** 4
**Preliminary Rating:** 4
**Recommendation:** Poster
**Final Rating:** 4

**Summary:**

The paper proposes an unpaired inpainting superresolution algorithm that extrapolates missing spine structures in axial and sagittal MRI images. Synthetic training pairs are generated to model the data shift and acquisition errors between sagittal slices and sagittal views of axial images. Overall, the paper presents a novel approach for improving spine MRI segmentation and processing, addressing challenges related to image resolution and acquisition errors.

**Strengths:**

- The paper proposes an unpaired inpainting superresolution algorithm that effectively extrapolates missing spine structures in axial and sagittal MRI images, addressing the challenge of partial volume effects and inter-vertebral movement during acquisition.

- Synthetic training pairs are generated to model the data shift and acquisition errors between sagittal slices and sagittal views of axial images, enabling the training of diffusion-based superresolution models on scaling factors larger than 6× without real paired data.

- The proposed method allows for automatic segmentation and processing of axial images, enabling the separation of individual vertebrae instances and facilitating vertebra-level labeling for atlas registration and large-scale studies.

**Weaknesses:**

- The paper does not provide a detailed analysis of the limitations or potential drawbacks of the proposed unpaired inpainting superresolution algorithm and its impact on the accuracy of the generated synthetic training pairs.

- The evaluation of the proposed method is limited to experimental benchmarks, and there is no comparison or analysis of the algorithm's performance on real clinical data or in comparison to other existing segmentation methods.

- The paper does not provide a comprehensive analysis of the potential impact of the synthetic training pairs and the various degradation functions on the overall segmentation accuracy and robustness of the proposed method.

**Detailed Comments:**

- Best to provide a detailed analysis of the limitations or potential drawbacks of the proposed unpaired inpainting superresolution algorithm and its impact on the accuracy of the generated synthetic training pairs.

- The evaluation of the proposed method is limited to experimental benchmarks, and please include the comparison or analysis of the algorithm's performance on real clinical data or in comparison to other existing segmentation methods.

- Please discuss the computational complexity or efficiency of the proposed algorithm.

- It is necessary to provide a comprehensive analysis of the potential impact of the synthetic training pairs and the various degradation functions on the overall segmentation accuracy and robustness of the proposed method.

**Justification Of Final Rating:**

Spine MRIs are often obtained in 2D axial or sagittal slices, which may not fully capture the vertebra structures. Aligning scans for multi-image superresolution is challenging due to volume effects and inter-vertebral movement during acquisition. This study introduces an unpaired inpainting superresolution algorithm that extrapolates missing spine structures. By applying the superresolution model to the z-dimension, it becomes possible to generate images that facilitate automatic segmentation and processing of axial images. The authors have satisfactorily addressed all of my concerns and I have no further comments.

**Justification Of The Preliminary Rating:**

The automatic segmentation and processing of axial images facilitated by the proposed method can have significant practical implications in clinical settings, improving the efficiency and accuracy of spinal cord analysis and diagnosis.

**Questions To Address In The Rebuttal:**

1. What are the limitations of the proposed algorithm?

2. How is the algorithm working on real clinical data?

3. What are the potential applications of the algorithm in clinical settings?

4. How does the proposed unpaired inpainting superresolution algorithm generate synthetic training pairs?

5. What are the multiple degradation functions used to model the data shift and acquisition errors?

6. How does diffusion-based superresolution outperform state-of-the-art models in the experimental results?

**Special Issue:**

No

---

> ### Author Response · Authors · 2024-03-15
>
> > The paper does not provide a detailed analysis of the limitations or potential drawbacks of the proposed unpaired inpainting superresolution algorithm and its impact on the accuracy of the generated synthetic training pairs. What are the limitations of the proposed algorithm?
>
> The large scaling factor compels the model to incorporate information not present in the original axial image, such as the thickness and hydration level of the intervertebral disk. Since the intervertebral disk can be smaller than a single axial slice, it's unrealistic to expect a perfect match with the real structure. To compensate for this, joint modeling of axial and sagittal images would be necessary, but we have not explored this possibility in this paper. With the segmentation and super-resolution capabilities now available, we can further refine our method. In the revision, we have included the statement: “SR for large scaling factors is strongly ill-posed, and our model is limited to a best guess. The structures between slices, as expected, vary from the real structure.”
>
> > The evaluation of the proposed method is limited to experimental benchmarks, and there is no comparison or analysis of the algorithm's performance on real clinical data or in comparison to other existing segmentation methods. And: How is the algorithm working on real clinical data?
>
> All evaluations are conducted using real clinical data collected from multiple sclerosis (MS) patients treated at clinic “Rechts der Isar” (under approval from the ethics board). Artificial pairs are generated solely for training purposes. Currently, there is no publicly available training dataset for axial segmentation, nor are there any pretrained segmentation models specifically tailored for axial images.
>
> > The paper does not provide a comprehensive analysis of the potential impact of the synthetic training pairs and the various degradation functions on the overall segmentation accuracy and robustness of the proposed method.
>
> The limitation of our analysis comes from the fact that the segmentation for axial images does not exist yet. With the SR we can generate these for the first time. All test metrics we use at least indirectly measure accuracy and robustness.
>
> Additionally, we ran the SR including segmentation on 360 subjects with two sessions for a follow up study. We found a labeling error in less than <1 % of the cases caused by the model.
>
> > What are the potential applications of the algorithm in clinical settings?
>
> Thank you for this important question. We feel there are many applications in a clinical context: We are able to conduct single image SR and segmentation on axial images. These are important for enabling segmentation, feature extraction and registration. These are prerequisites for automatic analysis for spine images, for instance in (1) longitudinal progression analysis of patients (2) radiomics and normative value analysis. For instance, the generated segmentations are used to register and compute the location of MS lesion on this cohort for a prospective study. Without this work, a manual annotation of all vertebra heights in all (2 x 360 Images) would have been required to do this study. Generating vertebra height labeling is time consuming and hard to obtain with this low resolution.
>
> > How does the proposed unpaired inpainting super resolution algorithm generate synthetic training pairs?
>
> Based on our proposed degradation scheme, we generate synthetic training pairs by degrading real sagittal images to resemble the appearance of a sagittal view of an axial image.
>
> > What are the multiple degradation functions used to model the data shift and acquisition errors?
>
> A SMORE degradation function is derived from two parameters: the input low resolution and the target high resolution. We varied the input resolution to maintain invariance to the starting resolution. Each sample utilizes only one SMORE filter.
>
> > How does diffusion-based super resolution outperform state-of-the-art models in the experimental results?
>
> Diffusion-based super resolution's superiority has been demonstrated in various studies, including our experiments detailed in table 1. The assumption is that diffusion is less prone to mode-collapse, better at representing the data distribution, and more responsive to conditioning.

---

> > ### Comment · Reviewer_HXjK · 2024-03-26
> >
> > The authors have addressed all of my concerns. I have no more comments.

---

### Official Review · Reviewer_H6eH · 2024-02-27

**Confidence:** 4
**Preliminary Rating:** 4
**Recommendation:** Poster
**Final Rating:** 4

**Summary:**

Existing challenges in multiple sclerosis (MS) studies: 1) Current Atlas-based registration tools require manual vertebra-level labels to function 2) Approaches based on whole spine segmentation for sagittal images lead to sub-optimal axial vertebra instance segmentation due to large slice thicknesses.
This paper proposes a super-resolution-driven solution that enhances the quality of axial images for segmentation to the exact resolution as the inplane resolution of a sagittal slice. The approach involves using sagittal images only for training, avoiding the difficulties associated with aligning sagittal and axial images.

The proposed approach improves the pre-existing degradation model SMORE based on scan properties to learn a prior that accommodates the distribution shifts between axial and sagittal view images.

**Strengths:**

The method is based on the recent and emerging trends in diffusion models.

The problem taken up by the authors is clinically essential. The method uses unpaired data, not necessitating ground truth images due to the thickly sliced spine images suffering from intrinsic motion, missing isotropic ground truth data, and misaligned sagittal volumes. These challenges are clearly shown in Figure 1.

**Weaknesses:**

Due to the large partial volume effects and inter-vertebral movement, we could not register the axial and sagittal image beyond the automatic alignment with an error of up to 30 mm. —-----------> A brief illustration of this challenge needs to be added at least in the appendix.

The breadth of comparative experiments could be better, meaning only two DL methods are compared. The authors have taken much of the allocated space for discussing degradations, squeezing the experiment section. To my knowledge, MIDL has seen manuscripts with extensive experiment sections so far.

The method is a direct application of the Palette with the degradation models inspired by SMORE. As such, the contributions of the work can be explicitly mentioned to improve the readability.

The authors mention the evaluation metric Betti error in most parts of the manuscript. I understand that they used Betti number error from the table captions. The referred paper has the metric Betti matching error, which clearly shows that Betti number error is indecisive where Dice scores are identical, even for a trivial case containing two digits (like MNIST). The evaluation of the efficacy of Betti number error on MNIST dataset with multiple datasets might give a better picture of how reliable this metric is.

**Detailed Comments:**

Introduction section - We change the zero-shot-learning approach of SMORE to an unpaired sagittal plain of axial to sagittal super-resolution. —--------> plain?? Typo??

The methods compared are only one GAN-based and a CNN, RCAN. The authors should also compare emerging transformer-based models in MRI super-resolution.

While most of the degradation models are simulated, the authors are encouraged to try out
https://github.com/Yonsei-MILab/MRI-Motion-Artifact-Simulation-Tool

The authors measure the segmentation performance of a pre-trained UNet on the different super-resolved axial images. —----> Please elaborate on this sentence

**Justification Of Final Rating:**

The authors have addressed most of my comments and have resolved the ambiguity with the Betti matching error metric. Therefore, I would like to stick to my original rating of "weak accept" and finalize it.

**Justification Of The Preliminary Rating:**

Their approach is interesting without target images for detail enhancement. There are high scopes to extend the work as a journal.
However, experiments and evaluation metrics can be expanded for further understanding.

**Questions To Address In The Rebuttal:**

There are also some questions in the weaknesses section.

In Section Related work -
The SMORE zero-shot approach is not suitable in our case due to the lack of rotation invariance and large-scale factors requiring the introduction of globally consistent information. —----> Can the authors rewrite this sentence related to “rotation invariance and large-scale factors” for clarity??

Is there no publicly available dataset for this, for instance, Zenodo spine for MRI?

Ablation studies look good, but because there are so many degradation models, which combinations are clinically relevant?? For example, SMORE one PSF and multi-PSF?

**Special Issue:**

Yes

---

> ### Author Response · Authors · 2024-03-15
>
> We thank the Reviewer for their feedback and comments.
>
> > A brief illustration of [partial volume effect, inter-vertebral movement and alignment error] challenge needs to be added at least in the appendix.
>
> Thank you for the valuable feedback, we agree that this improves the current manuscript. We have added an illustration (c.f. Fig. 7 in updated manuscript) indicating the mismatch in field of view (FOV) and exemplary misalignment.
>
> > The breadth of comparative experiments could be better, meaning only two DL methods are compared. The authors have taken much of the allocated space for discussing degradations, squeezing the experiment section.
>
> In order to highlight the main contribution of the paper, we deliberately limited the comparative study of different models because our contribution primarily lies in the degradation function and enabling SR under unpaired constraints.
>
> > The methods compared are only one GAN-based and a CNN, RCAN. The authors should also compare emerging transformer-based models in MRI super-resolution.
>
> Upon reviewer feedback, we have additionally tested HAT(arXiv:2205.04437) and PTNet(arXiv:2105.13993). Initially, both HAT and PTNet did not converge with the default hyperparameters. Subsequently, we modified HAT to use 4x4 blocks instead of 6x6 and achieved promising results. These have been included in Table1. Notably, training and inference are slower compared to our diffusion model.
>
> Vertebra dtr$\uparrow$  0.9805
>
> Betti $b_0$ er$\downarrow$ 0.098
>
> Betti $b_1$ er$\downarrow$ 0.240
>
> Axial Dice$\uparrow$ 0.681
>
> > The method is a direct application of the Palette with the degradation models inspired by SMORE. As such, the contributions of the work can be explicitly mentioned to improve readability. (plus, the typo of plain)
>
> Thanks for the hint. We've highlighted this as a final paragraph in the introduction to clarify the adaptation of SMORE's zero-shot learning approach and the use of the conditional diffusion model Palette.
>
> “We change the zero-shot-learning approach of SMORE to an unpaired sagittal **plane** of axial to sagittal super-resolution. We compare existing super-resolution networks **with the conditional diffusion models Palette.**”
>
> > The authors mention the evaluation metric Betti error in most parts of the manuscript. I understand that they used Betti number error from the table captions. […]
>
> Sorry for the ambiguity. We calculate a weak-form of Betti matching error by computing the error per-vertebra. $b_1$ serves as a lower-bound of the true matching error. $b_0$ is always measured correctly by construction.
>
> > [...] the authors are encouraged to try out MRI-Motion-Artifact-Simulation-Tool
>
> Thank you. We will explore this tool in the context of future work.
>
> > The authors measure the segmentation performance of a pre-trained UNet on the different super-resolved axial images. —----> Please elaborate on this sentence
>
> You're correct; the word "different" is inappropriate. We've simplified the sentence to " The axial SR is segmented by a pre-trained network." referring to a preliminary version from Möller et al. (2024).
>
> >  In Section Related work – […] Can the authors rewrite this sentence related to “rotation invariance and large-scale factors” for clarity?
>
> Thank you for the suggestion, we have revised the sentence accordingly (c.f. updated manuscript).
>
> > Is there no publicly available dataset for this, for instance, Zenodo spine for MRI?
>
> We are not aware of any publicly available dataset for axial data. Li et al. (2021) is the only paper we found, but their data is not accessible. For sagittal spine, Spider and MRSpineSeg Challenge datasets are available, but they exclusively focus on the lumbar spine. We aim to address the lack of annotation/segmentation nets for axial MR Spine. The ability to segment whole sagittal MRI spine has been only released to the public recently, specifically in Graf et al. (2023) and Möller et al. (2024). (citations c.f. paper).
>
> > Ablation studies look good, but because there are so many degradation models, which combinations are clinically relevant?
>
> Some degradation functions are intended to be invariant to changes in input images, while others should be more generalizable to different resolutions. All degradation functions are clinically relevant as they make the model generalize to a specific disturbance.
>
> > There are high scopes to extend the work as a journal. However, experiments and evaluation metrics can be expanded for further understanding.
>
> We agree and believe that extending the work to other MRI regions where isotropic ground truth exists would be valuable and enhance the results in future studies.

---

### Official Review · Reviewer_cFj9 · 2024-02-29

**Confidence:** 5
**Preliminary Rating:** 2
**Recommendation:** Poster
**Final Rating:** 2.5

**Summary:**

Authors present an image super-resolution method that focuses on
synthesizing missing slices in thick-slice axial MRI acquisitions of
the spine. Problems with segmentation and modeling with axial
acquisitions are highlighted. Alignment with sagittal acquisitions is
deemed problematic. To that end, a model that directly super-resolves
an axial acquisition in the through-plane direction is
proposed. Super-resolved images are evaluated via a segmentation. The
super-resolved images are segmented and then resampled back to the
coordinate frame of the original thick-slice axial
acquisition. Resampled segmentations are compared with ground
truth. Comparisons with different super-resolution techniques are also
presented alongside an ablation study.

**Strengths:**

- Authors do a sound modeling of the degradation process involved in
   acquiring axial images with thick-slices. I am assuming this
   technique can be applied to any type of axial MRI acquisition, not
   just spine.
- The article is well written, it is more or less clear what authors
   have done.
- High-resolution segmentation from thick-slice MRI, especially for
   the spine is a relevant problem.
- Authors use state-of-the-art conditional generation methods for
   synthesis.

**Weaknesses:**

- The work needs stronger motivation in my opinion.
  * The main claim of this article is that sagittal acquisitions
    cannot be used together with axial acquisitions. More precisely,
    registration of these two acquisitions does not work very
    well. This claim is not substantiated in this article. Examples of
    such misalignments, results of registration as well as details of
    the registration methods used should be provided.
  * The goal of the work is not very clear. Specifically, it is not
    clear whether authors are trying to replace sagittal
    acquisitions. If it is so, then stronger fidelity with the
    sagittal acquisitions need to be demonstrated. If the authors are
    not trying to replace sagittal acquisitions, then I am assuming
    one would always use the sagittal acquisitions for segmenting
    vertebrae as well as the discs for quantitative analysis. The
    axial acquisitions, as mentioned in the text, are preferred in the
    context of lesion activity. In this scenario, the need for
    segmentation of the axial acquisitions as well as why a
    segmentation obtained from the sagittal acquisitions could not be
    used for the axial acquisition should be justified.
- To the best of my understanding, the main novelty of the presented
  work is the degradation process. In terms of the learning algorithm
  the novelty is low. The conditional generation method is from a
  previous work.
- In the experimental analysis, comparing segmentations in the
  coordinate system of the axial-acquisition may not be evaluating the
  quality of the super-resolved image directly. I believe the
  comparison should be between a segmentation obtained from a
  super-resolved image and a sagittal acquisition.
- Authors compare all conditional generation methods using the
  degradation process proposed here. Therefore, the comparisons are
  actually focused towards showing performance gains due to
  Palette. However, this is not the contribution of this paper. The
  main contribution seems to be the degradation process, and that is
  the component that should be evaluated.

**Detailed Comments:**

In addition to my comments above, it is also unclear whether authors
do the degradation, synthesis as well as segmentation in 2D or in
3D. It would be very useful to clarify this part.

**Justification Of Final Rating:**

I thank the authors for the lengthy explanation. While I appreciate the applications more now, I am still not convinced regarding the demonstration of the novelty. The novelty - despite what authors claim - I believe is in the degradation process. As I wrote before, Palette is not new. Therefore, results in Table 1 only shows advantages of Palette - which is prior work. Furthermore, authors mention that the registration process fails and that leads to a mismatch. Dice score is claimed to not improve. It is crucial to report this DICE score and then compare it what is obtained after the SR process. Furthermore, Figure 7 provides some examples but after a simple alignment, do they not get better? Authors can provide segmentation overlays after alignment and compare with the result of the proposed method. In summary, I still believe (1) the novelty is in degradation and its value should be demonstrated more clearly and (2) a better demonstration of the difficulty of using the sagittal scans should be provided.

**Justification Of The Preliminary Rating:**

Please check the weaknesses I noted.
The motivation of the work is not very strong, the experimental analysis raises some concerns, and importantly the novelty is not very high. Given these, I am not very enthusiastic about the article in its current form. However, I believe the degradation process authors propose is interesting and can be demonstrated in a larger set of applications, which would be quite cool.

**Questions To Address In The Rebuttal:**

The main question to answer is the motivation. In its current format,
the motivation for the proposed work need to be substantially
strengthened.

Furthermore, I believe directly showing the benefits of the
degradation process, rather than comparing different conditional
generation methods would better position the article.

**Special Issue:**

No

---

> ### Author Response · Authors · 2024-03-15
>
> Thank you sincerely for your valuable feedback. We would like to address your concerns by showing you that our **motivation and goals** are multifaceted.
>
> **Improving Resolution:** Our aim is to enhance the resolution of spine images, particularly in non-isotropic scans. This is crucial because obtaining isotropic whole-spine images is often impractical. It's important to note that we're **not seeking to replace sagittal scans**. Both sagittal and axial images offer unique information, e.g. axial images better visualize facet joints, spinal cord lesions, nerve root ganglia and more. By enhancing both, we strive to achieve similar information content to that of an isotropic image. This improvement has multiple medical benefits, especially as the left-right coverage of sagittal scans is always limited: outer parts of the vertebra are never visible. Some large-scale studies like the UKBB lack sagittal images at all. As requested, we've included a new Figure 7 to illustrate axial and sagittal mismatch.
>
> **Axial Segmentation:** Currently, there's a lack of publicly available data or models for axial spine MRI. While Li et al. (2021) conducted semantic segmentation, their data is not public. Moreover, there's been no attempt at instance label segmentation. Our motivation for employing this approach is to generate segmentations for axial images based on the same principles used for sagittal MR and CT images.
>
> **Feature Extraction:** With our current methodology, we're able to provide semantic and instance segmentation for various downstream tasks. Since the vertebra structure is not sufficiently resolved in axial images, segmentation must be conducted in a super-resolved space. These feature points are invaluable e.g. for biomechanical modeling and point-based registration. We're now able to register axial images to the PAM50 atlas for comparative studies what was not possible before.
>
> In our introduction, we mention MS because the results described here are directly applicable to an upcoming study, where vertebral points of interest are needed to register axial images to an atlas, enabling us to assess the location of white and gray matter lesions along the spinal cord (refer to Fig. 5/6). Manual annotation in axial slices is challenging even for experienced radiologists, error-prone by the limited superior-inferior resolution, and time-consuming.
>
> We hope this clarifies our goals and motivations.
> > [....] the comparison should be between a segmentation obtained from a super-resolved image and a sagittal acquisition.
>
> While we agree that a comparison between segmentations of the SR image and the sag. T2w acquisition would be desirable, we cannot provide this comparison, as resampling would lose a lot of detail and we cannot align the sag. and axial SR images.
>
> > The main claim of this article is that sagittal acquisitions cannot be used together with axial acquisitions. […] This claim is not substantiated in this article.
>
> In discussions with our radiologist, registration experts, and other spine groups worldwide, we assessed the current status of deformable spine registration. Consensus was reached that spine registration remains largely unresolved in most cases without robust supervision such as segmentations or feature points, or significant manual intervention. We utilized various packages including Nipy, AirLab, ANTs, and SimpleITK for evaluation. Assessment was based on spinal cord Dice scores and visual alignment of vertebra corners. Unfortunately, registration either maintained the initial Dice score or deteriorated it. Despite attempting masking strategies for the spinal cord and incorporating spinal cord segmentation, alignment could not be improved due to significant discrepancies in pixel volumes. We also consulted multiple researchers regarding software capable of registering these images, but satisfactory results could not be achieved.
>
> > [...] the main novelty of the presented work is the degradation process. […]
>
> The main innovation lies in the degradation process and the methodology for training a SR model within our specified constraints. Joint unpaired and inpainting SR represent specialized cases that have received less attention in the medical super-resolution community. Additionally, our paper delves into the downstream tasks made possible by our constrained SR approach.
>
> > […] Therefore, the comparisons are actually focused towards showing performance gains due to Palette. […]
>
> We agree. Table 1 compares SR models to validate our choice of model. The performance gains are investigated in the experiments of Table 2.
>
> >[...] degradation, synthesis as well as segmentation in 2D or 3D?
>
> Apologies for the misunderstanding. We have clarified this in our manuscript. According to the physical principles of image acquisition, where we focused solely on typical 2D sequences commonly available for spine imaging, the degradation process had to be conducted in 2D. Meanwhile, the segmentation is 3D.

---

### Meta-Review · Area_Chair_eRao · 2024-04-04

**Recommendation:** Accept (Poster)
**Confidence:** 5

**Metareview:**

The  reviewers mostly agreed that the authors addressed their concerns and found novelty in some aspects of this paper. They agreed that there is a need to perform superresolution reconstruction to enable axial spine segmentation.

---

### Decision · Program_Chairs · 2024-04-05

Accept (Poster)